# Signal categorization by foraging animals depends on ecological diversity

David William Kikuchi[1]*, Anna Dornhaus[1], Vandana Gopeechund[2], Thomas N Sherratt[2]

[1]Department of Ecology and Evolutionary Biology, University of Arizona, Tucson, United States; [2]Department of Biology, Carleton University, Ottawa, Canada

**Abstract** Warning signals displayed by defended prey are mimicked by both mutualistic (Müllerian) and parasitic (Batesian) species. Yet mimicry is often imperfect: why does selection not improve mimicry? Predators create selection on warning signals, so predator psychology is crucial to understanding mimicry. We conducted experiments where humans acted as predators in a virtual ecosystem to ask how prey diversity affects the way that predators categorize prey phenotypes as profitable or unprofitable. The phenotypic diversity of prey communities strongly affected predator categorization. Higher diversity increased the likelihood that predators would use a 'key' trait to form broad categories, even if it meant committing errors. Broad categorization favors the evolution of mimicry. Both species richness and evenness contributed significantly to this effect. This lets us view the behavioral and evolutionary processes leading to mimicry in light of classical community ecology. Broad categorization by receivers is also likely to affect other forms of signaling.

DOI: https://doi.org/10.7554/eLife.43965.001

## Introduction

Signals between species can evolve whenever selection favors both the evolution of a signal display by a 'sender' species, and a response by a 'receiver' species (*Bradbury and Vehrencamp, 2011*; *Maynard Smith and Harper, 2003*). However, signal evolution is mediated not only by economics, but also by the psychology of receivers (*Endler and Basolo, 1998*; *Guilford and Dawkins, 1993*; *Rowe, 2013*; *Ryan et al., 1990*). In ecological communities, animals are faced with a diverse panoply of stimuli. How they categorize stimuli as worth responding to or ignoring will influence when and how signals evolve. Here, we explore how signal evolution is affected by the set of stimuli present in communities of different levels of complexity.

Warning signals are one of the best studied examples of interspecific communication – they advertise prey defenses to potential predators, reducing negative interactions for both predator and prey (*Wallace, 1867*). Mimicry occurs when warning signals coevolve among multiple prey species. Mimics can vary in their resemblance to models, with low fidelity ('imperfect') mimics representing something of a paradox (*Cuthill and TD, 1993*; *Dittrich et al., 1993*; *Kikuchi and Pfennig, 2013*; *Sherratt and Peet-Paré, 2017*). Understanding variation in the extent of mimicry is a problem that spans evolution, ecology, and cognitive psychology (*Guilford and Dawkins, 1993*; *Mallet, 2001*; *Rowe, 2013*; *Ruxton et al., 2018*), since selection on mimetic resemblance is mediated by the way that predators categorize prey as profitable or unprofitable (*Beatty et al., 2004*; *Gamberale-Stille et al., 2012*; *Getty, 1985*; *Ihalainen et al., 2012*; *Kazemi et al., 2014*; *Kikuchi and Sherratt, 2015*; *Oaten et al., 1975*; *Sherratt, 2002*; *Sherratt and Peet-Paré, 2017*; *Speed and Ruxton, 2010*).

The diversity of a community affects predator decisions about prey. For example, in an experiment with artificial prey, diversity affected how predators made decisions in response to

*For correspondence:
dwkikuchi@gmail.com

**Competing interests:** The authors declare that no competing interests exist.

warning signals that varied within a single, continuous dimension (*Ihalainen et al., 2012*). However, warning signals are often multicomponent, that is to say, complex – they consist of many different traits in concert (*Bradbury and Vehrencamp, 2011*; *Hebets and Papaj, 2005*; *Maynard Smith and Harper, 2003*). Indeed, genetic studies of mimicry complexes have revealed discrete variation among multiple mimetic traits (*Clarke and Sheppard, 1963*; *Jiggins, 2017*; *Kunte, 2009*; *Dasmahapatra et al., 2012*). Consequently, mimetic precision depends not only on how predators generalize within traits, but also on which traits they evaluate, and how they combine them to form higher-level categories. We use the terms 'categorization' and 'generalization' in the sense that categorization behavior results from using generalizations to make decisions (*Seger and Peterson, 2013*).

The use of 'key' traits is one simple way to classify prey (*Balogh and Leimar, 2005*; *Beatty et al., 2004*; *Gamberale-Stille et al., 2012*) – for example, using the rule 'avoid yellow prey' would mean that a predator would have to focus on the key trait of color (*Figure 1A*). However, the advantage of using key traits or any other form of categorization depends on the community in which these decisions are made (*Beatty et al., 2004*; *Ihalainen et al., 2012*; *Lindström et al., 2004*). In this study we examined the effects of communities on categorization, showing that different components of diversity have critical effects on which decision rules are used, and thus selection on mimetic signals.

## Results and discussion

The simplest, most widely studied component of diversity is species richness, the number of species found in a community (*Magurran, 1988*). It might be difficult for predators to identify and remember the properties of individual prey types in rich communities with a large variety of prey. Predators could be limited by memory capacity (*Beatty et al., 2004*; *MacDougall and Dawkins, 1998*), or by the substantial risks of sampling unfamiliar species of prey, some of which might be highly unprofitable to attack (*Cohen et al., 2007*; *Houston et al., 2012*; *Sherratt and Peet-Paré, 2017*). If predators do not remember the characteristics of each discrete prey type but instead use rules (such as avoid yellow), then they could reduce the difficulty of deciding what to eat in a rich community. Therefore, it has been hypothesized that, as richness increases, predators will be more likely to use a key trait to make decisions (*Beatty et al., 2004*; *Wilson et al., 2013*). Indeed, *Beatty et al. (2004)* found that predators *could* use a key trait to make decisions in diverse communities; however, in their experiment, if predators did not use the key trait, no discrimination was possible at all. To make strong inference that increased richness causes predators to use a key trait for decisions, it helps to include the choice to use either the key trait, or a specific, reliable trait that has more values (by values, we mean unique states or versions). This way, it is possible to determine if predators would actually switch their behavior if they did not have to.

We designed virtual prey communities where predators could either use a completely reliable trait that had many (2-8) different values to perfectly classify prey as 'good' or 'bad', or simplify decision-making by using an unreliable key trait (binary, with only two values) at the price of committing more errors (*Figure 1A*). In our virtual communities, prey always had two traits (color and shape – which one was reliable and which one was unreliable was randomized). Both traits were discrete, meaning that they could take on different values that did not grade continuously into one another. If the predator learned to identify good prey based on values of the reliable trait R, it could forage without errors. That is to say $P(good|R_i^+) = 1$, where + indicates that the value $R_i$ is positively correlated with profitability (for example, circle is always good in *Figure 1A*). The unreliable trait U was binary, having values of $U^+$ and $U^-$ (for example, prey each have one of two colors in *Figure 1A*). This binary trait only predicted whether an individual was 'good' with a probability of 0.78 (i.e. $P(good|U^+) = 0.78$). When only two values of each trait existed in the community (e.g. circle vs. star and blue vs. yellow; Experiment 1 in *Figure 1A*), the same number of individuals could be classified using either shape or color. However, using the unreliable trait (e.g. color) would carry the cost of committing more errors. Cognitive psychology experiments suggest that in this situation, the reliable trait will be used to the exclusion of the unreliable one due to a phenomenon called the relative validity effect (*Hall et al., 1977*; *Wagner et al., 1968*). In the relative validity effect, when an animal can learn to associate two cues with an outcome, it will learn to use the one that is more reliable (valid).

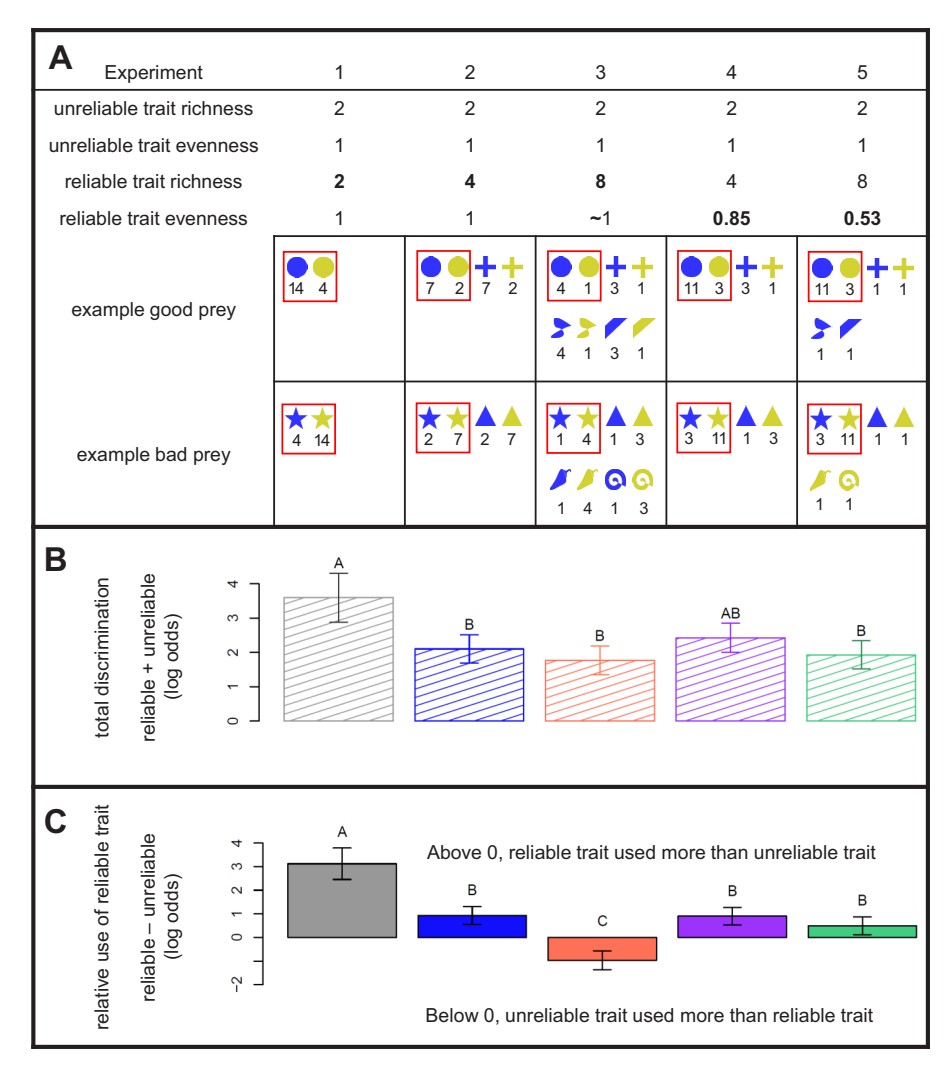

**Figure 1.** Design and results of Experiments 1 - 5. (**A**) Properties of the experimental prey communities used in this study, with examples. All communities had a 1:1 ratio of 'good' prey to 'bad' prey. A reliable trait allowed perfect discrimination. The richness and evenness of its values varied between experiments. An unreliable trait that did not vary between experiments yielded less accurate discrimination. The exact distribution of prey in each community is given below its richness and evenness statistics, with numbers to indicate the abundance of each prey. As drawn here, shape is the reliable trait (e.g. circle = good, star = bad), whereas color is the unreliable trait (blue = good 78% of the time, yellow = bad 78% of the time). Red boxes indicate the focal prey that were compared across experiments in panels B and C (their actual colors and shapes differed among treatments). (**B**) Total discrimination subjects exhibited towards focal prey, that is the summed influence of both reliable and unreliable traits. (**C**) Subjects' relative use of the reliable trait compared with the unreliable trait for discrimination, that is the difference between the effect of reliable and unreliable traits. The y-axis indicates the difference in the ability of the reliable trait to predict behavior compared to the unreliable trait. In (**B**) and (**C**), estimates are grouped using the Bonferroni correction for multiple pairwise comparisons, and 95% confidence intervals are shown. See Methods for details on interpreting log-odds.

DOI: https://doi.org/10.7554/eLife.43965.002

The following source data and figure supplements are available for figure 1:

**Source data 1.** Data used to generate *Figure 1* and its supplements.
DOI: https://doi.org/10.7554/eLife.43965.005
**Figure supplement 1.** Tabulated attack rates for prey of different types.
DOI: https://doi.org/10.7554/eLife.43965.003
**Figure supplement 2.** Tabulated attack rates for prey of different types.
DOI: https://doi.org/10.7554/eLife.43965.004

We recruited undergraduate student volunteers to serve as predators on our virtual prey communities. Each subject learned to forage on a grid of 36 prey during a training trial where they were allowed to attack up to 18 of the prey, and received feedback on whether each was 'good' or 'bad' to eat in the form of a smiley face with a chirp or an X with a gong sound. Their 'life bar' would also rise or fall accordingly (subjects lost twice as much life for attacking 'bad' prey than they gained for eating 'good' prey). After subjects finished the training trial, they took a test trial where they could choose as many prey as they liked, but received no feedback (*Figure 2*). The test trial served two purposes: 1) it allowed us to measure subjects' categorization behavior without changing it by providing feedback, and 2) because the test trial was always the same, it allowed us to compare subjects' categorization behavior after foraging in different training communities. Subjects participated in five experiments presented in random order (*Figure 2*; *Figure 2—figure supplement 1*). In Experiment 1, our control to see if the relative validity effect held with our design, we found that the reliable trait was used almost exclusively (*Supplementary file 1*).

We tested three mutually exclusive hypotheses for how predators will classify their prey as its phenotypic richness increases. Predators had to choose how much to rely on the key trait $U$ at the price of committing some errors, or the completely reliable trait $R$ at the price of learning about and memorizing multiple values. The first hypothesis was that they should select the former when the price of information (e.g. memory, costs of exploration) limits the profitability of using the reliable trait, so that as richness increases, they should use the unreliable trait to a greater degree (e.g. use color more and begin to ignore shape as the richness of shapes increases; *Figure 3A*). The second hypothesis was that if the relative validity effect were an invariant aspect of predator psychology, then predators should persist in using the reliable trait across different levels of richness (e.g., learn all of the shapes across Experiments 1–3, always ignoring color; *Figure 3B*). Indeed, associative learning experiments on the relative validity effect do not show a difference in which trait is used as the number of its values increases – that is, subjects always use the most reliable trait (*Baetu et al., 2005*; *Murphy et al., 2001*). However, the number of trait values in these experiments has been low. A third hypothesis is that at high levels of richness, predators may not be able to parse all of the information that they are confronted with and will guess randomly with respect to the reliable and unreliable traits (*Figure 3C*).

To test these hypotheses, we performed two experiments (2 and 3) that had higher richness than Experiment 1. Subjects used the reliable and/or unreliable traits to make decisions in all experiments (*Figure 1B*), which allowed us to reject the hypothesis that they would not use either trait at high diversities. In Experiment 2, where there were four values of the reliable trait (two associated with profitability, two with unprofitability), subjects decreased their use of the reliable trait

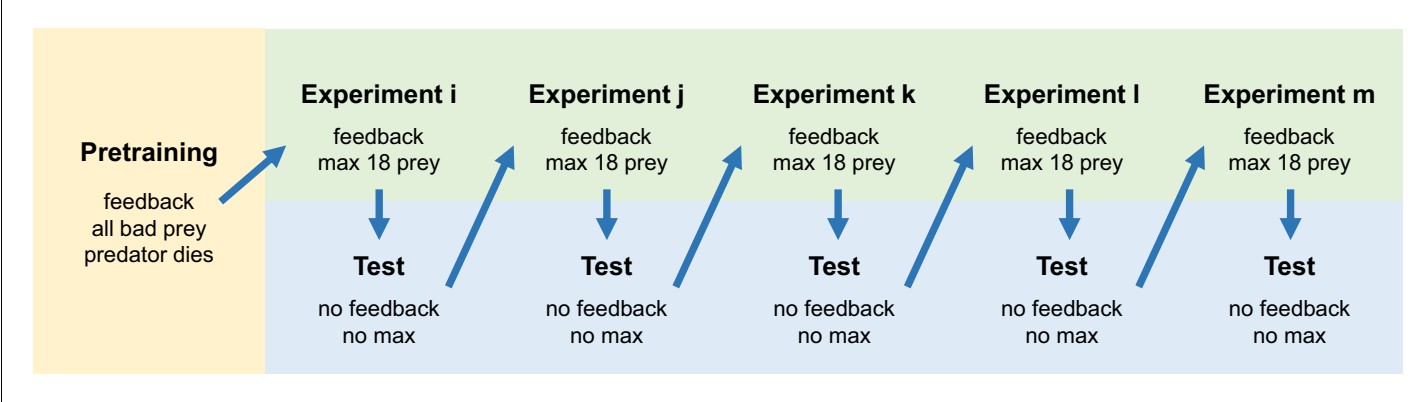

**Figure 2.** General experimental procedures. Subjects were randomly assigned to one of four different treatments within each of five experiments. Between treatments, colors and shapes were shuffled with respect to 'good' and 'bad' prey to prevent subjects from generalizing across experiments.

DOI: https://doi.org/10.7554/eLife.43965.006

The following figure supplement is available for figure 2:

**Figure supplement 1.** All experimental treatments used in this study.

DOI: https://doi.org/10.7554/eLife.43965.007

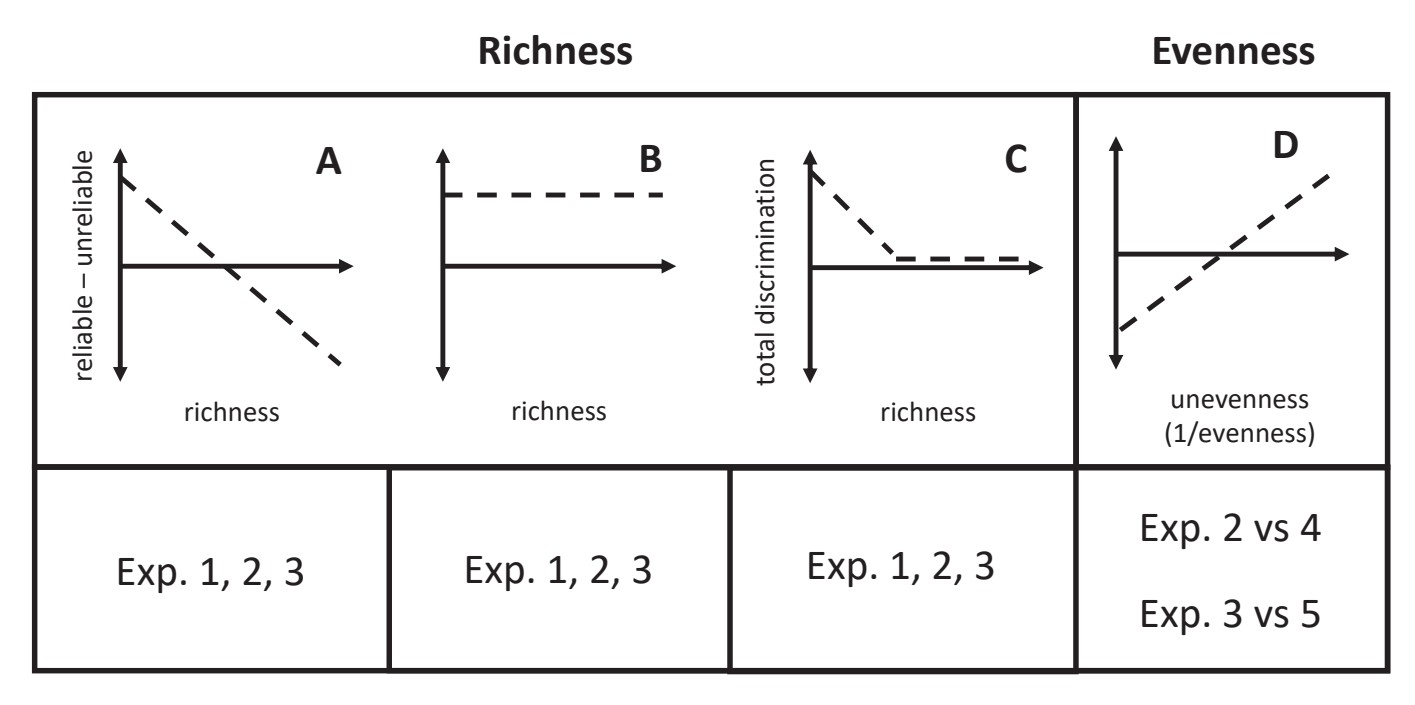

**Figure 3.** Graphical predictions of hypotheses described in the text. (A) The relative use of the reliable trait will decrease if increased prey richness causes predators to prefer the unreliable trait. (B) If the relative validity effect is robust to changes in richness, predators will always use the reliable trait. (C) If predators cannot process all of the information available in diverse communities, they will guess randomly. (D) If the reduced effective richness of prey in uneven communities reduces the costs of information, then use of the reliable trait will increase.

DOI: https://doi.org/10.7554/eLife.43965.008

significantly (*Figure 1C*). In Experiment 3, where there were eight values of the reliable trait, subjects again significantly decreased their use of the reliable trait - in fact, they used the unreliable trait more (*Figure 1C*). These results allow us to reject the hypothesis that the relative validity effect is constant across levels of richness. Instead, they support the hypothesis that species-rich communities carry a high price of information, either in memory constraints or the risks of acquiring information, which increases predators' tendency to use the simpler (binary) yet unreliable trait in decision-making. This is strong evidence supporting the hypothesis that in rich communities, mimicry could evolve easily on the basis of key features that predators use for identification (*Beatty et al., 2004*). It also supports important theoretical models of how mimicry evolves that depend upon key features (*Balogh et al., 2010*; *Balogh and Leimar, 2005*; *Gamberale-Stille et al., 2012*). Furthermore, this result is critical to the stability of warning signals that are parasitized by Batesian mimics because it implies that predators will not immediately switch to using more reliable traits simply because they are available. High species richness could still favor broad categorization.

The other component of community diversity is species evenness, or relative abundance (*Magurran, 1988*; *Tuomisto, 2012*). In every community, some species are common, while other species are rare. This 'overrepresentation' of some species and underrepresentation of others reduces the effective number of species in the community (i.e., lower evenness means effectively fewer species; *Jost, 2010*). Consequently, it is reasonable to hypothesize that unevenness will decrease predators' use of the key trait to form categories, reversing the effect of increasing richness. In fact, prior work has shown that both higher frequencies of profitable, non-mimetic prey and lower frequencies of Batesian mimics relaxes selection on mimicry (*Finkbeiner et al., 2018*; *Harper and Pfennig, 2007*; *Iserbyt et al., 2011*; *Lindström et al., 2004*; *Lindström et al., 1997*; *Pfennig et al., 2001*). Here, we ask more generally about the effects of evenness per se, where there is a distribution of relative abundance within both profitable and unprofitable prey.

In uneven communities, rarer prey types will be less important food resources. This led us to test the hypothesis that unevenness will decrease predators' tendency to categorize prey using an

unreliable key trait. It predicts that in uneven communities, predators will use the reliable trait more than in an evenly distributed community with the same phenotypic richness (*Figure 3D*).

To test this hypothesis, we conducted two experiments with uneven communities. Experiment 4 featured the same eight phenotypes as Experiment 2, and Experiment 5 involved the precisely the same shapes and colors as Experiment 3, but in Experiments 4 and 5, one of the focal 'good' values and one of the focal 'bad' values of prey were much more abundant than the others. The reliable trait was used to a significantly greater degree in Experiment 5 compared to Experiment 3, as evinced by their placement in different *post-hoc* groupings (*Figure 1C*). However, we observed no difference in the use of the reliable trait between Experiments 2 and 4 (*Figure 1C*).

The hypothesis was rejected; unevenness increased subjects' use of the reliable trait, but only at high levels of richness. We attribute this outcome to a simple cause. In both Experiments 4 and 5, where communities were uneven, predators mainly distinguished between the most abundant profitable value of the reliable trait, and all others. In other words, predators mainly attacked prey with the most abundant 'good' value, and categorized the other values as not worth attacking. Effectively, the most abundant, good, reliable value became a preferred alternative prey. In support of this interpretation, a model that included two different values for 'good' prey and two different values for 'bad' prey within the reliable trait $R$ fit significantly better than one that lumped 'good' values together, and 'bad' values together (likelihood ratio test, $\chi^2_{24} = 70.1$, $P < 0.001$). Predators attacked the abundant good prey much more than any other kind of prey in Experiment 5 and exhibited this behavior to a lesser degree in Experiment 4 (*Figure 1—figure supplement 1*).

Results from the uneven communities contrast with the pattern from Experiments 1–3, which shows increasing reliance on the unreliable trait with increasing richness. We suggest that the unevenness of a prey community will be negatively correlated with predators' reliance upon key traits to form categories. This may make mimicry less likely to evolve in uneven communities. Furthermore, it connects evenness, a fundamental parameter of community ecology, to the concept of alternative prey from mimicry theory: when one species of profitable, non-mimetic prey is relatively abundant, selection on other, rarer prey to evolve mimicry will be relaxed (*Getty, 1985*; *Holling, 1965*; *Ihalainen et al., 2012*; *Kokko et al., 2003*; *Lindström et al., 2004*).

Very few studies from natural systems have collated the data that would be required to measure the relationship between community diversity and signaling systems. *Wilson et al. (2013)* argued that a negative relationship between mimetic precision and community diversity stems from increased generalization by predators in more diverse communities of velvet ants. Additionally, in experimentally manipulated communities of flowering plants, increased color diversity tended to increase visitation rates by pollinating insects (*Fornoff et al., 2017*). It would be interesting to know if this occurred because individual pollinators relied on coarser phenotypic categories in richer communities.

Our hypotheses might also be applicable to subsets of communities. Particularly, specialist predators might experience smaller prey communities than generalist predators, and specialist pollinators might visit fewer species of flowers than generalist pollinators. For example, different mimicry rings of *Heliconius* butterflies are segregated by microhabitat (*Elias et al., 2008*), and exposed to different suites of predators as a consequence; predation favors precise mimics within their preferred microhabitats (*Willmott et al., 2017*). Habitat specialization could reduce the size of the community about which a particular bird must learn, allowing them to select for precise mimicry (or none at all) because coarse categorization based on key traits would not occur.

Theoretical models suggest that other ecological conditions than those we explored here can also affect the number of traits that predators use to make decisions. Under some circumstances when the costs of attacking 'bad' prey are in a particular balance with the benefits of attacking 'good' prey, trusting only the most reliable trait may be most adaptive (*Rubi and Stephens, 2016*). Yet changing the cost:benefit ratio or underlying frequency of good prey can favor using multiple traits, or using no trait at all (*Sherratt and Holen, 2018*).

## Conclusions

Both prey richness and evenness affected predator categorization behavior. The result that richness will favor mimicry because predators use a key trait in categorization, even at the cost of decisional

accuracy, was not anticipated by literature on associative learning. Generally, our results show that ecological diversity predicts both the origins and maintenance of mimicry.

Our results are relevant to problems in signal evolution beyond defensive mimicry, for all communication systems evolve within the context of communities. Floral phenotypes have long been remarked to appear similar to one another, presumably to signal to shared pollinators (*Ollerton et al., 2009*; *Schaefer and Ruxton, 2010*; *Schaefer and Ruxton, 2009*; *Shrestha et al., 2013*). It follows from our study that in richer communities, flowers could benefit from sharing key traits that allow pollinators to categorize them together. Likewise, other signals between species such as pursuit deterrence (*Caro, 1995*), mobbing calls (*Magrath et al., 2015*), herbivore-induced plant volatiles (*Schuman and Baldwin, 2016*), and ripening fruit (*Willson and Whelan, 1990*) may also evolve to share key traits in rich, even communities. Very few studies have collated the data that would be required to measure the relationship between community context and signaling systems. Yet clearly, communities have the potential to produce striking effects on signal evolution.

## Materials and methods

### Experiments

Human volunteers can readily be recruited to participate in short computer games that are completely harmless and yield large quantities of data. Experiments of similar design performed by other animals and humans have yielded qualitatively similar results (*Alatalo and Mappes, 1996* vs. *Beatty et al., 2005*; *Cuthill et al., 2005* vs. *Fraser et al., 2007*; *Kazemi et al., 2014* vs. *Sherratt et al., 2015*), although humans often learn faster. A major exception to this is XOR tasks (also called correlated features tasks), where no single dimension is helpful for categorization, but their combinations are. Humans rank XOR tasks as easier than several other kinds of tasks, but other primates rank them as more difficult (*Smith et al., 2004*). Differences between humans and non-humans in this task are thought to be mediated by language (*Smith et al., 2011*). Importantly, our experiments did not include XOR tasks, so human behavior is more likely to be representative of non-human species.

We designed a computer game in Psychtoolbox-3 for Matlab R2017b where subjects were asked to hunt for artificial prey (*Brainard, 1997*; *Kleiner et al., 2007*). The general format of the game was presented to subjects during the pre-training trial, which only occurred once at the very beginning of the game. Subjects saw a grid of 36 prey that constituted a prey community. They were told that they could sample as many as 18 of those prey (half), or terminate sampling prematurely. However, in the pretraining trial they did not have the option to stop sampling early. This forced them to participate long enough to understand the basics. When they attacked a 'bad' prey, it was immediately covered with an X while a gong noise played and the screen froze for two seconds. In addition, a life bar on the side of the screen would decrease. If subjects attacked a 'good' prey, it would be covered with a smiley face while a chirp sounded, no time penalty occurred, and the life bar immediately increased. The amount of life lost from attacking 'bad' prey was twice that gained from attacking 'good' prey, but the life bar did not change over time unless subjects attacked, that is they had no penalty for moving slowly or quickly. In the pretraining trial, all prey were 'bad', and the subject's life bar was set low to start so that they would die after attacking four prey. When they died, a lonesome whistle sounded and the screen froze for two seconds. We arranged the pretraining trial in this way so that all subjects would begin the experiments from the same starting point, and be more likely to pay attention to avoiding 'bad' prey in addition to finding 'good' prey.

Once subjects had completed pretraining, they were told that they would now proceed to the real game, which was the first experimental training trial. Subjects were randomly assigned to an experiment, and within that experiment, randomly assigned to a treatment. Treatments were designed so that within each experiment, shape would be the reliable trait in two treatments, and color the reliable trait in the other two. Which colors and shapes were associated with 'good' and 'bad' prey were permuted within the treatments in which they were the reliable trait. The full array of treatments is presented in *Figure 2—figure supplement 1*. Subjects began the game with their life bars at half full so that they could encounter eight 'bad' prey in a row before dying. This was intended to give them motivation to forage yet made it unlikely that they would actually 'die'. If they did die, the training trial ended, but the flow of the game did not otherwise change.

Once subjects completed the training trial, they were told that they had the opportunity to play a bonus round (the test trial) to increase their score. They were told that they could attack as many prey as they liked in the bonus round, but that they could stop whenever they wanted to. They were also told that they would receive absolutely no feedback until the bonus round was over. We designed the test trial without feedback so that subjects would not continue to learn (and hence change the categories they had formed) during the test trial. The prey in the test trial were always the same no matter what experimental treatment subjects experienced. These test prey always included prey with four values of the reliable trait (two good, two bad). After subjects finished the test trial, they were told their total score across both rounds. This was calculated as the sum of all 'good' prey attacked less all 'bad' prey attacked, but subjects were not informed of this formula – the only purpose of telling them these scores was to keep them motivated in both the training and test trials. By making it difficult to tell exactly how the score was calculated, though, we encouraged them to focus on their life bar during training trials, which continued to fluctuate with a cost:benefit ratio of 2:1 for bad:good prey.

Subjects experienced each of the five experiments in random order, taking the test trial immediately after completing each one (*Figure 2*). The only exceptions to this were a few subjects that completed fewer than five experiments to improve the balance of our design. A pseudo-random design might have made this unnecessary, but we did not want to unintentionally induce any bias in the order in which treatments were presented.

When subjects had completed all five experiments and the respective test trials, they were asked to take a color blindness test (Ishihara plates 6, 8, 13, and 23). This did not constitute a medically professional diagnosis of color blindness, so they were not informed of their results, but any subject failing the test was excluded from the final dataset. In total, we recruited 45 volunteers who passed this basic test from the Carleton University Student Union in Ottawa, Canada.

## Data analysis

We designed our analysis of the test trial to answer the question of how well the reliable trait predicted subjects' behavior in each experiment, relative to the unreliable trait. This was critical to testing the predictions of the first two hypotheses about species richness (*Figure 3A & B*), and the hypothesis about evenness (*Figure 3D*).

There were three steps to this analysis: the first was to find, in each experiment, the estimated effects of the reliable and unreliable traits on subjects' decisions to attack or reject prey. We estimated these effects with a statistical model of subjects' decisions. Second, we found the *difference* between the effects of the reliable and unreliable traits in each experiment. Third, we performed pairwise comparisons of these differences between experiments. This required finding the differences of differences. Both of these difference calculations used parameter estimates that we obtained from the statistical model. We describe our methods below, and also refer readers to the RMarkdown in *Supplementary file 1*.

It is necessary to understand the structure of the data. We analyzed subjects' attacks on focal prey, defined as the four most abundant prey present in the training trial (e.g. red boxes in *Figure 1A*). The focal prey were also always present in the test trial. The test trial included some trait values that were not present in the training – we eliminated them from analysis. To describe the effects of the reliable trait, which differed in its number of values between experiments, we recoded the values of the focal prey according to whether they were associated with profitability or unprofitabilty during training. That is to say, we combined $R_i^+$ and $R_i^-$ into just two values, R$^+$ and R$^-$. Taking an example from *Figure 1A*, circle and cross in Experiments 2 – 5 were recoded as 'G', and star and triangle were recoded as 'B'. The end result of recoding was that the reliable and unreliable traits could be analyzed as factors with only two values, making them both binary (*Supplementary file 1*). This made their relative contributions easy to compare by simply looking at their effect sizes once they had been centered and incorporated into a suitable statistical model (*Schielzeth, 2010*).

We fit a model that was designed to find the effects of the reliable and unreliable traits and their standard errors, rather than to fit our data as well as possible. To do this, we fit the model without an intercept, and without main effects of the reliable trait and unreliable trait. This violates the principle of marginality, but our aim was not to test hypotheses with the model. Excluding the intercept and two main effects allowed us to directly find the effects of interest, instead of having to perform additional calculations (*Schielzeth, 2010*). We took into account potential confounding variables in

constructing our model, however, as they could affect estimates of the effects. Therefore, we included interactions with order, once it had been centered. Centering causes the estimates of lower-order terms to be made at the mean value of a variable, so that lower-order terms can be interpreted independently of interactions (*Schielzeth, 2010*). We also included subject identity as a main effect to control for variation among individuals (models that included it as a random effect did not converge). In R pseudocode, the model that we fit is shown below, with the specific quantities of interest in bold:

```
glm((attacked, not attacked) ~ 0 + experiment + subject ID + unreliable trait:
experiment + reliable trait:experiment + unreliable trait:experiment:order +
reliable trait:experiment:order
```

with a logit link function. All data are available in *Figure 1—source data 1*.

The model provided the estimates of the effects of the reliable and unreliable traits in each experiment, with standard errors. We used these estimates to find their differences. The difference was appropriate because log-odds were the units for effect size estimates from the model (since it was fit with a logit link). Log-odds are an ideal metric of trait importance to decision-making because they are easily converted into the odds of attack for different kinds of prey. If $\beta_{R1}$ is the log-odds estimate for $P(\text{attack}|R^+) - P(\text{attack}|R^-)$ and $\beta_{U1}$ is the log-odds estimate for $P(\text{attack}|U^+) - P(\text{attack}|U^-)$ in Experiment 1, then $\exp(\beta_{R1}-\beta_{U1})$ gives the relative difference odds of attack due to the reliable trait compared to the unreliable trait in Experiment 1. For example, in Experiment 1 the effect size of the 'good' value of the reliable trait is 3.36, and the corresponding estimate for the unreliable trait is 0.24. This means $R^+$ prey have $e^{3.36} = 28.8$ times the odds of attack compared with prey that are $R^-$, but $U^+$ prey only suffer an increase in odds of attack of $e^{0.24} = 1.27$ compared to $U^-$ prey. Finding their difference as $3.36 - 0.24 = 3.12$ means that in Experiment 1, $R^+$ prey have $e^{3.12} = 22.6$ times the attack risk of $U^+$ prey.

A function to find the difference in the effect of the reliable and unreliable traits in each experiment is simple subtraction (e.g., $R_1$-$U_1$). However, finding the standard errors of the difference is more complicated. The delta method is one way of approximating the standard error of a function of estimated effects (*Bolker, 2008*). To implement the delta method, we used the function deltaMethod from the 'car' package in R 3.4.4 (see *Supplementary file 1*; *Fox et al., 2018*). This completed the first difference calculation.

We again used the delta method to perform pairwise comparisons between the relative importance of traits between experiments, using a Bonferroni correction for multiple comparisons. This allowed us to answer our question of how the use of the reliable versus unreliable traits changed across experiments. The statistical significance of our results depended on whether or not numerically calculated confidence intervals for the difference between estimates included zero or not, which is displayed in the groupings in *Figure 1B and C*.

We repeated the operations described above to find the difference of the *sums* of the effects of both traits, which is a way of describing the total discrimination of subjects in an experiment. We did this to test the prediction that high species richness results in random guessing (*Figure 3C*).

Finally, we tested the prediction that in uneven communities, predators would focus on the most abundant good prey. We modeled subjects' attack decisions just as we did above, but instead of using combined 'good' and 'bad' values of the reliable trait, we used the original focal prey values, for example circle, cross, star, triangle. Thus, there were up to two 'good' and two 'bad' values of the reliable trait. We used the likelihood ratio test to compare this model to the one we fit above. A significant difference in model fit would mean that subjects treated different values of good and/or bad prey differently, which is the qualitative pattern we observed in Experiments 4 and 5 (*Figure 1—figure supplement 2*). Note that although both models violated marginality, this does not matter for the comparison of fit that we performed between them.

## Acknowledgements

We thank members of the Dornhaus and Sherratt labs for helpful comments and support. We thank members of Bob Wilson's lab for advice on coding in Matlab. We thank our reviewers, including Bernhard Schmid, for very helpful comments. Human subjects research was carried out with the permission of the Carleton University Research Ethics Board-B under permit number 13385 14–0276. Funding: DWK was funded by NIH-K12GM000708. TNS was funded by an NSERC Discovery Grant.

## Additional information

### Funding

| Funder | Grant reference number | Author |
|---|---|---|
| National Institutes of Health | K12GM000708 | David William Kikuchi |
| Natural Sciences and Engineering Research Council of Canada | | Thomas N Sherratt |

The funders had no role in study design, data collection and interpretation, or the decision to submit the work for publication.

### Author contributions
David William Kikuchi, Conceptualization, Software, Investigation, Visualization, Methodology, Writing—original draft, Writing—review and editing; Anna Dornhaus, Writing—review and editing; Vandana Gopeechund, Data curation; Thomas N Sherratt, Conceptualization, Writing—review and editing

### Author ORCIDs
David William Kikuchi http://orcid.org/0000-0002-7379-2788

### Ethics
Human subjects: Consent process is described in the Methods. Human subjects research was carried out with the permission of the Carleton University Research Ethics Board-B under permit number 13385 14-0276.

### Decision letter and Author response
Decision letter https://doi.org/10.7554/eLife.43965.012
Author response https://doi.org/10.7554/eLife.43965.013

## Additional files

### Supplementary files
• Supplementary file 1. RMarkdown with full analysis, including code to reproduce all results. The file includes a legend to the columns of the *Figure 1—source data 1* with detailed explanation of the variable codings. It is preferable to use this file as a guide to the data.
DOI: https://doi.org/10.7554/eLife.43965.009
• Transparent reporting form
DOI: https://doi.org/10.7554/eLife.43965.010

### Data availability
All data for this study are present in the supporting files, and source code to produce the figures from those files is included in the Supplementary RMarkdown file.

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
