## [Decision Letter]

Thank you for submitting your article "Community diversity determines predator generalization in mimicry" for consideration by *eLife*. Your article has been reviewed by three peer reviewers, including Bernhard Schmid as the Reviewing Editor, and the evaluation has been overseen by Ian Baldwin as the Senior Editor.

The reviewers have discussed the reviews with one another and the Reviewing Editor has drafted this decision to help you prepare a revised submission.

Summary:

This paper asks how generalist predators should choose their prey in species-rich prey communities if prey species are either fully palatable or fully unpalatable. The expectation is that trait generalization by predators should occur even at the expense of making some mistakes in choosing prey, because this increases foraging efficiency. The authors suggest that as a consequence of this expectation less perfect warning signals would be sufficient to deter predators in species-rich as compared with species-poor prey communities. As a further consequence, this might select for imperfect mimicry in diverse prey communities.

The initial expectation is tested with human subjects choosing mock prey species on a computer, of which 50% or species are good and 50% are bad, and all have a reliable (not generalized) and an unreliable (generalized) trait. The results generally confirm that in more diverse prey communities subjects switch from reliable to unreliable traits for identification. They also show that it is effective richness that matters, that is, species-rich but uneven communities still allow subjects to use the reliable trait of the abundant good prey species. Interestingly, subjects were better at learning to choose the good prey than at avoiding the bad prey. As a corollary, the theory may be more appropriate for mutualistic interactions such as between plants and pollinators, where generalization should lead to the evolution of similar flower traits between co-flowering species.

Essential revisions:

The reviewers identified three major areas that should be addressed during revisions. First, the scope of the paper is more focused than suggested by the title and Introduction. Second, the description of the statistical analysis is too sketchy and would not allow one to repeat the analysis. Third, the interpretation of the results should include a comparison with empirical data and discuss potential extensions and alternatives.

1) Scope: it seems misleading to focus on mimicry in the title and Introduction, because the paper is not really about selection of imperfect mimicry. Please choose a title that reflects your main results. Using humans as predators is clever, but still you need more justification (or mention a caveat) about your assumption that psychological decisions translate to natural predator behavior.

2) Statistics: the description of the data analysis is very hard to follow (especially the first, third and last paragraphs in the subsection “Data Analysis”). It is therefore hard to assess the appropriateness of the approach. It is important that you provide data and code so readers can do their own calculations. Generalized linear models with random terms can yield very different results depending on the specific method used. For example it can be dangerous to use readily available functions from *R* such as glmer, try at least a comparison with glm and constructing the correct approximate F-ratios from the mean deviance changes (make sure scale is estimated and not set to 1) (see e.g. textbook by McCullagh and Nelder). Also, weren't reliable and unreliable traits both present within each experiment? If so, how could there ever be an 'absence' of one of these traits (subsection “Data Analysis”, second paragraph)? There is also some concern why you are removing 'order' from the analyses. Why not partition out its effects, but remain focused on interpreting your main hypotheses? You can still interpret lower-order effects in a model, even if higher-order interactions are present if contrasts are properly coded (for an example, see Schielzeth, 2010). It appears that the statistical analyses deal with the data from the test trials of the different experiments (Figure 2). However, this is not explicitly stated in the manuscript.

3) Results and Discussion: first of all, consider renaming Results as 'Results and Discussion' and renaming Discussion as 'Conclusions'. This seems like a more logical grouping based on how the manuscript is written.

3a) Comparison with empirical data: please discuss to what extent imperfect mimicry is especially prominent in species-rich prey communities. Imperfect mimicry definitely occurs in species-poor prey communities, e.g. at high and low latitudes, whereas striking cases of highly perfected mimicry can be found in very species-rich communities. As an illustration, mimicry was discovered by Bates and Mueller through their perceived need for an explanation of striking similarity in appearance of certain butterfly species in very species-rich communities. Also, please look for publications/empirical data from observational studies and biodiversity experiments (e.g. German Exploratories or Jena Experiment and Cedar Creek biodiversity experiments, respectively) to compare with your predictions.

3b) Potential extensions and alternatives: if you want to keep the setting of co-evolution between prey and predators in diverse communities, you should expand the discussion to other possibilities besides generalists avoiding bad prey based on signal generalization. There is a potentially important aspect missing in your logic, namely that of competition between predators, which you should discuss. Usually, biodiversity experiments and observational studies find that with the number of prey species the number of predator species also increases. This is then tentatively explained by specialization (predator niche differentiation), which becomes possible with high prey diversity. Thus, while prey diversity may select for generalization in generalist predators, it may instead also select for the generalist predators to become more specialized on particular prey, thus making the need to identify bad prey obsolete, as indeed in part observed in your experiments. Provide more background about situations that require predators to negatively select out unpalatable prey as opposed to positively select palatable prey!

---

## [Author Response]

Essential revisions:The reviewers identified three major areas that should be addressed during revisions. First, the scope of the paper is more focused than suggested by the title and Introduction. Second, the description of the statistical analysis is too sketchy and would not allow one to repeat the analysis. Third, the interpretation of the results should include a comparison with empirical data and discuss potential extensions and alternatives.1) Scope: it seems misleading to focus on mimicry in the title and Introduction, because the paper is not really about selection of imperfect mimicry. Please choose a title that reflects your main results. Using humans as predators is clever, but still you need more justification (or mention a caveat) about your assumption that psychological decisions translate to natural predator behavior.

We have changed our title to read “Signal categorization by foraging animals depends on ecological diversity” which is more in keeping with our main results. We have also reworked the first part of our Introduction to make it more general:

“Signals between species can evolve whenever selection favors both the evolution of a signal display by a “sender” species, and a response by a “receiver” species (Bradbury and Vehrencamp, 2011; Maynard Smith and Harper, 2003). […] Here, we explore how signal evolution is affected by the set of stimuli present in communities of different levels of complexity.”

We have added a paragraph explaining our rationale behind our use of human subjects:

“Human volunteers can readily be recruited to participate in short computer games that are completely harmless and yield large quantities of data. […] Importantly, our experiments did not include XOR tasks, so human behavior is more likely to be representative of non-human species.”

2) Statistics: the description of the data analysis is very hard to follow (especially the first, third and last paragraphs in the subsection “Data Analysis”). It is therefore hard to assess the appropriateness of the approach. It is important that you provide data and code so readers can do their own calculations. Generalized linear models with random terms can yield very different results depending on the specific method used. For example it can be dangerous to use readily available functions from R such as glmer, try at least a comparison with glm and constructing the correct approximate F-ratios from the mean deviance changes (make sure scale is estimated and not set to 1) (see e.g. textbook by McCullagh and Nelder). Also, weren't reliable and unreliable traits both present within each experiment? If so, how could there ever be an 'absence' of one of these traits (subsection “Data Analysis”, second paragraph)? There is also some concern why you are removing 'order' from the analyses. Why not partition out its effects, but remain focused on interpreting your main hypotheses? You can still interpret lower-order effects in a model, even if higher-order interactions are present if contrasts are properly coded (for an example, see Schielzeth 2010). It appears that the statistical analyses deal with the data from the test trials of the different experiments (Figure 2). However, this is not explicitly stated in the manuscript.

We apologise that our code was not visible to the reviewers, although it was uploaded with our submission. In this revision, our analysis is available as an RMarkdown document (Supplementary file 1).

The analyses we have conducted in the revision were done with glm, as glmer failed to converge with the additional interactions requested.

Our new analysis now includes order, using centered variables so we can interpret the estimates of interest even in the presence of a significant interaction with order (please see revised Materials and methods).

We have modified the third paragraph of the subsection “Data Analysis” so that it is clear we mean different factor levels of each trait.

Thank you for referring us to Schielzeth, 2010. We have used centering to repeat our analysis with higher-level interactions, while retaining the ability to analyze lower-level effects.

We have specified that the test trial was analyzed.

3) Results and Discussion: first of all, consider renaming Results as 'Results and Discussion' and renaming Discussion as 'Conclusions'. This seems like a more logical grouping based on how the manuscript is written.

This is very sensible, we have made this change.

3a) Comparison with empirical data: please discuss to what extent imperfect mimicry is especially prominent in species-rich prey communities. Imperfect mimicry definitely occurs in species-poor prey communities, e.g. at high and low latitudes, whereas striking cases of highly perfected mimicry can be found in very species-rich communities. As an illustration, mimicry was discovered by Bates and Mueller through their perceived need for an explanation of striking similarity in appearance of certain butterfly species in very species-rich communities. Also, please look for publications/empirical data from observational studies and biodiversity experiments (e.g. German Exploratories or Jena Experiment and Cedar Creek biodiversity experiments, respectively) to compare with your predictions.

We have explored the literature to find such data, but very little is available. Certainly there are examples of poor mimicry at high latitudes, and precise mimicry at low latitudes, but currently data do not exist to test whether these are outliers, or the norm. We have suggested empirical studies worth more investigation at the end of the Results and Discussion, which include one from the Jena Experiment:

“Very few studies from natural systems have collated the data that would be required to measure the relationship between community diversity and signaling systems. […] It would be interesting to know if this occurred because individual pollinators relied on coarser phenotypic categories in richer communities.”

3b) Potential extensions and alternatives: if you want to keep the setting of co-evolution between prey and predators in diverse communities, you should expand the discussion to other possibilities besides generalists avoiding bad prey based on signal generalization. There is a potentially important aspect missing in your logic, namely that of competition between predators, which you should discuss. Usually, biodiversity experiments and observational studies find that with the number of prey species the number of predator species also increases. This is then tentatively explained by specialization (predator niche differentiation), which becomes possible with high prey diversity. Thus, while prey diversity may select for generalization in generalist predators, it may instead also select for the generalist predators to become more specialized on particular prey, thus making the need to identify bad prey obsolete, as indeed in part observed in your experiments. Provide more background about situations that require predators to negatively select out unpalatable prey as opposed to positively select palatable prey!

We are circumspect about competition between predators driving prey specialization directly; rich communities often have predator species that partition niche space spatially or temporally rather than by particular prey types (e.g. in species-rich Amazonia, some birds follow army ants, and some forage on individual territories). So it is not obvious how predator diversity would change our results. Nevertheless, the reviewers’ point is very valuable and well taken: not all predators will experience the full diversity of a community. We have elaborated on the potential importance of this work to specialist or generalist predators:

“Our hypotheses might also be applicable to subsets of communities. […] Habitat specialization could reduce the size of the community about which a particular bird must learn, allowing them to select for precise mimicry (or none at all) because coarse categorization based on key traits would not occur.”